# Pathophysiology of Acute Kidney Injury in Malaria and Non-Malarial Febrile Illness: A Prospective Cohort Study

**DOI:** 10.3390/pathogens11040436

**Published:** 2022-04-03

**Authors:** Michael T. Hawkes, Aleksandra Leligdowicz, Anthony Batte, Geoffrey Situma, Kathleen Zhong, Sophie Namasopo, Robert O. Opoka, Kevin C. Kain, Andrea L. Conroy

**Affiliations:** 1Division of Pediatric Infectious Diseases, University of Alberta, Edmonton, AB T6G 2R3, Canada; mthawkes@ualberta.ca; 2Division of Critical Care Medicine, Robarts Research Institute, University of Western Ontario, London, ON N6A 5A5, Canada; aleligdo@uwo.ca; 3Child Health and Development Center, Makerere University College of Health Sciences, Kampala, Uganda; abatte2002@yahoo.com; 4CHILD Biomedical Research Laboratory, Global Health Uganda, Kampala, Uganda; gsituma123@gmail.com; 5Sandra Rotman Centre for Global Health, Toronto General Hospital, University Health Network, Toronto, ON M5G 1L7, Canada; kzhong@uhnres.utoronto.ca (K.Z.); kevin.kain@uhn.ca (K.C.K.); 6Department of Medicine, University of Toronto, Toronto, ON M5G 1L7, Canada; 7Kabale District Hospital, Kabale, Uganda; snamasopo@gmail.com; 8Department of Paediatrics and Child Health, Makerere University, Kampala, Uganda; opokabob@yahoo.com; 9Ryan White Center for Pediatric Infectious Disease and Global Health, Center for Global Health, Indiana University School of Medicine, Indianapolis, IN 46202, USA

**Keywords:** acute kidney injury, malaria, non-malarial febrile illness, sepsis, mortality, acute infection, children, sub-Saharan Africa, immune activation, endothelial activation

## Abstract

Acute kidney injury (AKI) is a life-threatening complication. Malaria and sepsis are leading causes of AKI in low-and-middle-income countries, but its etiology and pathogenesis are poorly understood. A prospective observational cohort study was conducted to evaluate pathways of immune and endothelial activation in children hospitalized with an acute febrile illness in Uganda. The relationship between clinical outcome and AKI, defined using the Kidney Disease: Improving Global Outcomes criteria, was investigated. The study included 967 participants (mean age 1.67 years, 44.7% female) with 687 (71.0%) positive for malaria by rapid diagnostic test and 280 (29.1%) children had a non-malarial febrile illness (NMFI). The frequency of AKI was higher in children with NMFI compared to malaria (AKI, 55.0% vs. 46.7%, *p* = 0.02). However, the frequency of severe AKI (stage 2 or 3 AKI) was comparable (12.1% vs. 10.5%, *p* = 0.45). Circulating markers of both immune and endothelial activation were associated with severe AKI. Children who had malaria and AKI had increased mortality (no AKI, 0.8% vs. AKI, 4.1%, *p* = 0.005), while there was no difference in mortality among children with NMFI (no AKI, 4.0% vs. AKI, 4.6%, *p* = 0.81). AKI is a common complication in children hospitalized with acute infections. Immune and endothelial activation appear to play central roles in the pathogenesis of AKI.

## 1. Introduction

Globally, 85% of acute kidney injury (AKI) cases occur in low-and-middle-income countries (LMIC) [1]. In sub-Saharan Africa, malaria and sepsis are leading causes of AKI in children [2,3], with estimates of AKI prevalence ranging from 24 to 59% in children with severe malaria using consensus definitions [4,5,6,7,8,9]. AKI is associated with increased risk of in-hospital and post-discharge mortality [5,10,11,12]. In addition, AKI is a risk factor for chronic kidney disease (CKD) [5,11,13,14,15,16] as well as non-renal morbidity in survivors [4,5,17,18]. Among severe malaria survivors, AKI is an independent risk factor for long-term neurocognitive deficits and behavioral problems [5,19]. Given the global burden of AKI and CKD, and the unknown impact of AKI during childhood on health outcomes across an individual’s lifespan, globally representative studies are needed to understand the etiology and pathophysiology of AKI across diverse populations and settings.

In the context of infection, the host response is an important determinant of disease severity and survival [20,21]. Endothelial and immune activation represent two pathways associated with disease severity and mortality and are well-described in sepsis-associated AKI [22,23,24]. These pathways can be interrogated by measuring levels of circulating biomarkers [25,26,27]. In African children, *Plasmodium falciparum* is the primary cause of severe malaria and is characterized by the cytoadherence of parasitized red blood cells to the endothelium [28,29]. Parasite sequestration can exacerbate microvascular dysfunction contributing to tissue hypoxia and endothelial activation [28]. Severe malaria-associated AKI is associated with higher sequestered parasite biomass and endothelial activation compared to children without AKI [5,30]. However, the contribution of immune and endothelial activation in children with AKI in the context of malaria or non-malarial febrile illnesses (NMFI) has not been systematically evaluated.

In the present study, we characterize pathways of immune and endothelial activation in hospitalized children < 5 years of age with an acute febrile illness. We hypothesize that children with severe malaria-associated AKI experience greater endothelial and immune activation compared to children with NMFI-associated AKI.

## 2. Results

The analysis included 967 hospitalized children enrolled between February 2012 and August 2013 (Figure 1). The mean (SD) age at admission was 1.67 years (1.07) and 44.7% of participants were female. Overall, 71.0% of children tested positive for malaria. Participant characteristics based on malaria status are described in Table 1.

### 2.1. Prevalence of AKI in Malaria vs. NMFI

The prevalence of AKI at admission was 49.1%. AKI was observed in 46.7% of children with malaria and 55.0% in children with a non-malaria febrile illness (NMFI) (*p* = 0.020). Malaria was associated with 38% reduced odds of AKI compared to children with a NMFI with a odds ratio (OR) of 0.62 (95% CI 0.46 to 0.83) after adjusting for child age, sex, and disease severity (*p* = 0.001). The prevalence of severe AKI was comparable in children with malaria vs. NMFI at 12.1% and 10.5%, respectively (*p* = 0.45). Differences in participant characteristics based on malaria status in children with or without severe AKI are presented in Table 2. Jaundice, a clinical sign of hemolysis, was associated with severe AKI in patients with malaria (OR 3.65 (95% CI 2.08 to 6.41), *p* < 0.0001) and NMFI (OR 5.06 (95% CI 1.71 to 14.97), *p* = 0.003). Delayed capillary refill time, a clinical sign of shock, was associated with severe AKI in malaria (OR 1.91 (95% CI 1.04 to 3.51), *p* = 0.04) but did not reach statistical significance among children with NMFI (OR 1.79 (95% CI 0.68 to 4.72), *p* = 0.24).

We quantified levels of cystatin C as an alternate functional biomarker of AKI in children at enrollment. AKI represents an abrupt loss of kidney function due to changes in kidney filtration (assessed using filtration markers such as creatinine or cystatin C) or urine output. Cystatin C levels had moderate ability to discriminate between children with or without AKI in malaria with an area under the receiver operator characteristic curve (AUC) of 0.71 (95% CI 0.64 to 0.77) and good discriminatory ability in children with NMFI with an AUC of 0.82 (95% CI 0.72 to 0.91). Using a pre-established cystatin C cut-off of >0.8 mg/L [31], 19.4% of children were biomarker positive for AKI. The frequency of positive cystatin C was 16.7% in children with malaria compared to 26.1% in children with a NMFI (*p* < 0.001). Furthermore, among children with malaria, a positive cystatin C test was associated with 3.59-fold increased odds of severe AKI (95% CI 2.12 to 6.09) compared to an 11.46-fold increase in the odds of severe AKI (95% CI 5.02 to 26.14) in children with a NMFI.

### 2.2. Immune and Endothelial Activation in AKI in Children with Malaria vs. NMFI

To study the pathophysiology of severe AKI in children based on the etiology of infection, we quantified biomarkers involved in the host’s response to infection (Figure 2). Immune activation was assessed using levels of C-X-C motif chemokine Ligand 10 (CXCL10)/interferon γ-induced protein 10 kDa (IP-10), chitinase-3-like protein 1 (CHI3L1), soluble tumour necrosis factor receptor-1 (sTNFR1), soluble triggering receptor expressed on myelocytes (sTREM-1), interleukin 6 (IL-6), and interleukin 8 (IL-8). Endothelial activation was assessed using angiopoietin-2 (Angpt-2), angiopoietin-1 (Angpt1), soluble fms-like tyrosine kinase-1 (sFlt-1), soluble vascular cell adhesion molecules (VCAM-1), and soluble intercellular adhesion molecule-1 (sICAM-1).

In children with both malaria and NMFI, there was evidence of endothelial activation (increased Angpt-2 and sFlt-1), as well as immune activation (increased sTNFR1, CHI3L1 and sTREM-1) in children with severe AKI (Figure 2, adjusted *p* < 0.05 for all). 

However, among children with malaria, there was evidence of more pronounced endothelial activation with increases in both sVCAM-1 and sICAM-1 and higher IL-8 levels. To evaluate potential interactions between AKI and infection-mediated immune and endothelial activation, we standardized biomarker levels so the relative change could be compared across fever etiology (malaria vs. NMFI) and AKI status (no severe AKI vs. severe AKI) (Figure 3). When evaluating levels of sFlt-1, there was a significant interaction term between malaria and AKI, suggesting that AKI and malaria interact to synergistically increase sFlt-1 levels. 

In children with malaria and NMFI, there was an increase in CHI3L1, sTFNR1, sTREM-1, and Angpt-2 in the presence of severe AKI (Figure 3). However, there were some differences between the nature of the host response in children based on the presence of malaria. Among children with malaria and severe AKI, there was evidence of more endothelial activation compared to children with NMFI and severe AKI. Conversely, children with NMFI and severe AKI had higher levels of cystatin C and IL-8 compared to children with malaria-associated severe AKI. Biomarkers of immune activation were correlated with each other, as were biomarkers of endothelial activation (Figure 3). Using factor analysis as a data reduction technique, we computed composite indices of immune and endothelial activation (IAI and EAI), based on a linear combination of biomarker levels from each panel (Figure 3). The EAI and IAI were correlated with the LOD score as a marker of disease severity (rho = 0.323 (EAI), rho = 0.308 (IAI), *p* < 0.0001 for both), and were higher in fatal cases (*p* < 0.0001 for both), providing evidence of convergent validity of the scales. Higher IAI and EAI were associated with severe AKI in both malaria and NMFI (Figure 3).

### 2.3. Relationship between AKI and Mortality

Overall, 2.9% of study participants died in-hospital. We evaluated the relationship between AKI, severe AKI, and mortality among children based on infection etiology. Children who had malaria and AKI had increased mortality (no AKI, 0.8% vs. AKI, 4.1%, *p* = 0.005), while there was no difference in mortality among children with NMFI (no AKI, 4.0% vs. AKI, 4.6%, *p* = 0.81). A similar relationship was seen in children with severe AKI where children who had malaria and severe AKI had increased mortality (Stage 1 AKI, 1.6% vs. severe AKI, 12.5%, *p* < 0.001), while the difference in mortality among children with NMFI was not significant (Stage 1 AKI, 3.3% vs. severe AKI, 8.8%, *p* = 0.18). In models adjusting for age, sex, and disease severity, the presence of severe AKI showed a stronger relationship with mortality in children with malaria compared to children with NFMI (malaria, risk ratio (RR) 2.73 (95% CI 0.94 to 7.90), *p* = 0.06; NMFI, RR 1.04 (95% CI 0.34 to 3.18), *p* = 0.94). Similarly, when positive cystatin C was used as the diagnostic marker for AKI, there was an association with increased risk of mortality among children with malaria (RR 2.97 (95% CI 1.29 to 6.87), *p* = 0.01) but not among children with NMFI (RR 1.83 (95% CI 0.77 to 4.30), *p* = 0.17) after adjusting for age and sex and severity of illness.

## 3. Discussion

In the present study, AKI was common in children hospitalized for malaria and NMFI. We identified several biomarkers of immune (IL-8, sTNFR1, sTREM-1, CHI3L1) and endothelial (Angpt-2, sFlt-1) activation associated with AKI in children with both malaria and NMFI. In this population, AKI was associated with more pronounced endothelial activation and a higher risk of mortality in children with malaria, but not among children with NMFI.

The mechanisms of AKI in febrile Ugandan children are likely multifactorial. Jaundice may be a clinical sign of hemolysis and was associated with AKI in both malaria and NMFI patients in this study. Hemolysis is common with *P. falciparum*, an intraerythrocytic parasite, and results in the release of free heme, a known nephrotoxin [32]. Delayed capillary refill, a clinical sign of poor tissue perfusion and shock, which may indicate pre-renal causes of AKI, was associated with AKI in malaria patients but not in children with NMFI. Host biomarkers provide additional mechanistic insights. Cystatin C is an established functional biomarker of AKI in pediatric populations. Overall, 19.4% of the participants had a positive cystatin C test using a cut-off established in a pediatric AKI cohort [31]. Children with NMFI were more likely to have a positive cystatin C test than children with severe malaria, suggesting that children with a NMFI may have a greater reduction in kidney function compared to children with severe malaria. However, the relationship between severe AKI and mortality was more evident in the context of severe malaria, independent of disease severity, suggesting other pathways may be contributing to increased mortality in children with severe malaria and AKI.

Endothelial activation appears to be a common pathway leading to AKI in both malaria and NMFI but may be more accentuated in malaria-associated AKI. Increased endothelial activation in the context of malaria-associated AKI may reflect direct injury to the glomerular endothelium due to *P. falciparum* cytoadherence [33]. However, as severe malaria is a multi-system disease, we cannot rule out endothelial activation in other organ systems contributing to enhanced systemic endothelial activation in the context of severe malaria-associated AKI. In the present study, we demonstrated increased endothelial activation in severe malaria-associated AKI with increased circulating levels of cell surface molecules sICAM-1, sVCAM-1, and sFlt-1. Irrespective of infection etiology, plasma Angpt-2 levels were elevated in severe AKI. These results are consistent with Angpt-2 as a marker of disease severity in critical illness [4,25,26,27,30,34,35,36]. Angpt-2 levels have been implicated in both severe malaria and sepsis-associated AKI [30,37]. A polymorphism (rs2920656C > T) near the *ANGPT2* gene associated with a functional decrease in Angpt-2 production is associated with reduced risk of developing a sub-phenotype of AKI defined based on sTNFR1, Angpt-2, and Angpt-1 [38]. Changes in sTNFR1, Angpt-2, and Angpt-1 were all evident in severe malaria-associated AKI and future studies are warranted to evaluate whether polymorphisms in the *ANGPT2* gene may also modify the risk of AKI severity and recovery in the context of severe malaria.

In the present study, severe AKI was also characterized by immune activation with specific elevations in sTNFR1, sTREM-1, CHI3L1, and IL-8. A higher immune activation index (IAI) was associated with severe AKI in both malaria and NMFI. These findings are consistent with markers of immune activation as biomarkers of AKI and disease severity [39], to risk-stratify patients at risk of mortality in outpatient settings [40], and for prognostic enrichment of patients with sepsis [41]. The current study lends additional support to CHI3L1 and sTREM-1 as biomarkers of AKI [42], disease severity [25,43], and mortality in severe malaria [26,44]. Additional longitudinal studies are needed to understand the utility of these biomarkers in relation to AKI duration, severity, and kidney recovery. There is increasing evidence of long-term non-renal complications following pediatric AKI, including neurocognitive and behavioral problems in survivors [19], cardiac dysfunction [45], and altered growth. Identification of biomarkers that can facilitate early AKI identification, implementation of kidney-protective care, and assessment of long-term risk of renal and non-renal complications of AKI have the potential to transform kidney care in children, particularly in resource limited settings [46].

To evaluate immune activation and endothelial activation as concepts, we used confirmatory factor analysis to derive indices of immune and endothelial activation based on panels of correlated biomarkers with similar biological functions (Figure 3). The IAI gave roughly equal weighting to inflammatory cytokines IL-6 and IL-8, and neutrophil activation markers CHI3L1, sTNFR-1, and sTREM-1, but lower weighting to the chemokine CXCL-10 (Figure 3). The EAI gave nearly equal weighting to endothelial cell surface molecules sFlt-1, sVCAM-1, and sICAM-1, vasoactive molecule Angpt-2, and negative weighting to Angpt-1, consistent with its role in promoting endothelial quiescence (opposite in biological effect to the markers of endothelial activation) (Figure 3) [34]. Taken together, the IAI and EAI appear to be measuring unified constructs consistent with known roles of the selected biomarkers. 

The strengths of this study include the large sample size which enabled investigation of the relationships between AKI severity, immune and endothelial activation, and mortality in children hospitalized with an acute febrile illness due to malaria and NMFI. Limitations of this study include the lack of information on the etiology of febrile illness in children without malaria and a lack of data on co-infections. As most children tested positive for malaria, the study was underpowered to detect smaller differences in AKI-related mortality in children with NMFI. Despite these limitations, we show distinct differences in markers of life-threatening host response to infection in children with malaria-associated AKI compared to children with AKI due to other causes and demonstrate that children with malaria are particularly susceptible to AKI-related mortality. 

Together, these data suggest that pathways of immune and endothelial activation underlie the pathophysiology of AKI in children with both severe malaria-associated AKI and AKI related to NMFI. However, endothelial activation may be more pronounced in the context of severe malaria than NMFI-associated AKI. Further, endothelial activation may explain– in part– worse clinical outcomes in children with severe malaria and AKI. Additional studies are needed to delineate differences in AKI etiology and pathophysiology using urine biomarkers to differentiate between functional and structural kidney injury. Finally, studies are needed to evaluate pathways of adaptive vs. maladaptive repair following AKI in children with malaria vs. NMFI to identify opportunities to improve kidney recovery among children with severe malaria-associated AKI.

## 4. Materials and Methods

### 4.1. Study Population

This study was nested within a previously described prospective cohort study of children hospitalized with an acute febrile illness at Jinja Regional Referral Hospital in Uganda [44,47]. Children were eligible if they were between two months to five years of age; had a history of fever within 48 h, or axillary temperature greater than 37.5 °C; required hospitalization according to the admitting physician’s judgment; and the parent/guardian consented to participate in the study. Exclusion criteria included diarrheal illness without other symptoms. Malaria was assessed using a three-band rapid diagnostic test (RDT) with *P. falciparum* histidine-rich protein 2 (HRP2) and pan-malaria lactate dehydrogenase (pLDH) (First Response Malaria Ag. HRP2/pLDH Combo Rapid Diagnostic Test, Premier Medical Corporation Limited, Mumbai, India) [48].

### 4.2. Study Design

Within the parent study, we conducted a nested sub-study to evaluate differences in the host response associated with AKI in children with malaria and a non-malarial febrile illness (NMFI). As a secondary outcome, we examined the association between AKI and in-hospital mortality in children based on malaria status. A sample size calculation indicated that we would need a minimum of 715 patients to detect a difference in mortality between patients with and without AKI at admission, assuming a baseline mortality rate of 4% [47], a prevalence of AKI at admission of 33%, and a relative risk of death associated with AKI of 2.5 or greater, at the α = 0.05 level of confidence, with 80% power.

### 4.3. Defining Acute Kidney Injury

AKI was defined using the Kidney Disease: Improving Global Outcomes (KDIGO) criteria based on a 1.5-fold increase in serum creatinine from estimated baseline and staged as follows: stage 1, 1.5–1.9-fold increase in creatinine over baseline; stage 2, 2.0–2.9-fold increase over baseline; stage 3 ≥ 3.0-fold increase over baseline. Baseline creatinine was estimated using a height-independent approach, assuming a GFR of 120 mL/min per 1.73 m^2^ as described [49]. AKI was classified as severe if it was stage 2 or 3 [50]. Creatinine was tested using the modified Jaffe colorimetric method on an Alinity c instrument (Abbott, Lake Forest, IL, USA), which is traceable to an isotope dilution mass spectrometry (IDMS) reference method. Cystatin C was included as an alternative functional marker of AKI measured by Luminex (R&D Systems, Minneapolis, MN, USA) and classified as positive if levels were > 0.8 mg/L [31].

### 4.4. Measurement of Biomarkers of Host Response to Infection

Two panels of biomarkers were designed to interrogate host pathways of: (1) immune activation; and (2) endothelial activation. Selected molecules have been previously validated as biologically plausible and clinically informative markers host response to infection. Immune activation was assessed using circulating (plasma) levels of C-X-C motif chemokine Ligand 10 (CXCL10)/interferon γ-induced protein 10 kDa (IP-10) [25,26], chitinase-3-like protein 1 (CHI3L1) [42], soluble tumor necrosis factor receptor-1 (sTNFR1) [41], soluble triggering receptor expressed on myelocytes (sTREM-1) [40], interleukin 6 (IL-6), and interleukin 8 (IL-8) [41]. Endothelial activation was assessed using angiopoietin-2 (Angpt-2) [34,35], angiopoietin-1 (Angpt1) [27], soluble fms-like tyrosine kinase-1 (sFlt-1) [30], soluble vascular cell adhesion molecules (VCAM-1) [30], and soluble intercellular adhesion molecule-1 (sICAM-1) [30]. Biomarkers of immune and endothelial activation were evaluated using a custom Magnetic Luminex^®^ Performance Assay (R&D Systems) in EDTA anticoagulated plasma stored at −80 °C until testing [51].

### 4.5. Statistical Analysis

Data were analyzed using STATAv14.0 and GraphPad Prism v7.0. Continuous data were analyzed using Student’s *t*-test or Wilcoxon rank-sum test. Categorical data were analyzed using Pearson’s Chi-square test. To evaluate the relationship between clinical signs and severe AKI, logistic regression was used. To evaluate biomarker signatures across groups, biomarker concentrations were standardized to have a mean of 0 and a standard deviation of 1. The mean standardized concentration is presented by AKI and malaria status and differences analyzed using linear regression was used to evaluate differences in biomarker levels based on the presence of severe AKI and malaria. The reference category was children without malaria or severe AKI and the p values from malaria, severe AKI and the interaction term were used to evaluate significance. To adjust for multiple comparisons, the Benjamini–Hochberg procedure was used, adjusting for 36 comparisons. The relationships between biomarkers were evaluated using the non-parametric Spearman correlation. Confirmatory factor analysis was used to derive a single latent construct of immune activation (immune activation index, IAI) and endothelial activation (endothelial activation index, EAI). Differences in the indices were evaluated in children based on malaria and severe AKI status using a one-way ANOVA with Sidak’s multiple comparison test. To evaluate the relationship between AKI and a positive cystatin C test and mortality, Poisson regression was used with robust variance estimates and models adjusted for participant age, sex, and disease severity (assessed using LODS).

## 5. Conclusions

AKI in children hospitalized with severe malaria is associated with greater endothelial activation and higher mortality. With the International Society of Nephrology goal of eliminating preventable deaths due to AKI by 2025 (0 by 25 initiative), and efforts to reduce and eliminate malaria, increased collaborative research efforts directed towards the study of malaria-associated AKI are urgently needed.

## Figures and Tables

**Figure 1 pathogens-11-00436-f001:**
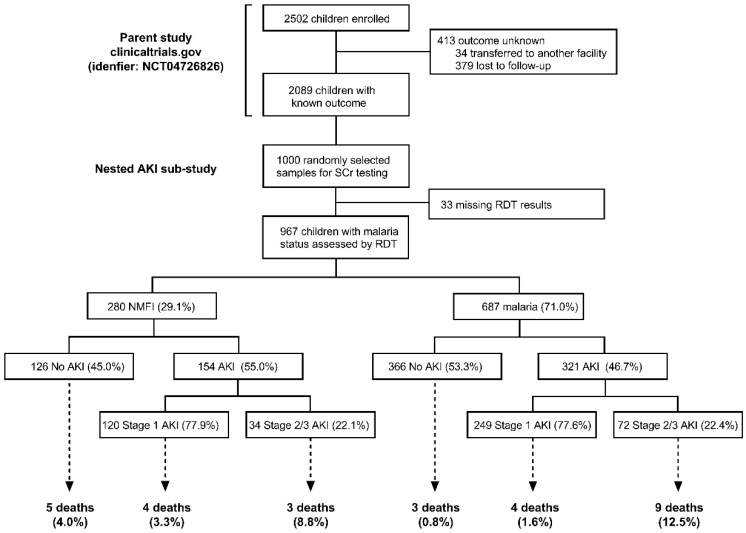
Flowchart of the study population.

**Figure 2 pathogens-11-00436-f002:**
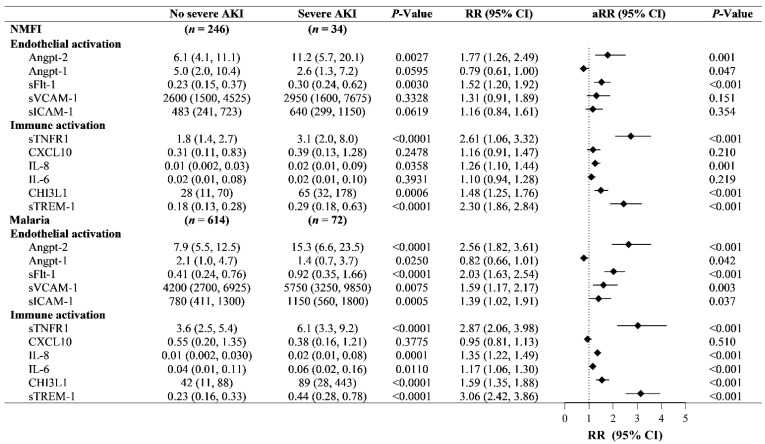
**Forest plot biomarkers of endothelial and immune activation in children with severe AKI in the context of malaria and non-malarial febrile illness (NMFI).** Biomarker levels presented as median (IQR) based on AKI status. Risk ratio (RR) and 95% confidence intervals (CI) calculated using Poisson regression with robust variance estimates on log_10_ biomarker levels adjusting (aRR) for participant age and sex. Following adjustment for 22 comparisons, a *p* < 0.002 is considered significant.

**Figure 3 pathogens-11-00436-f003:**
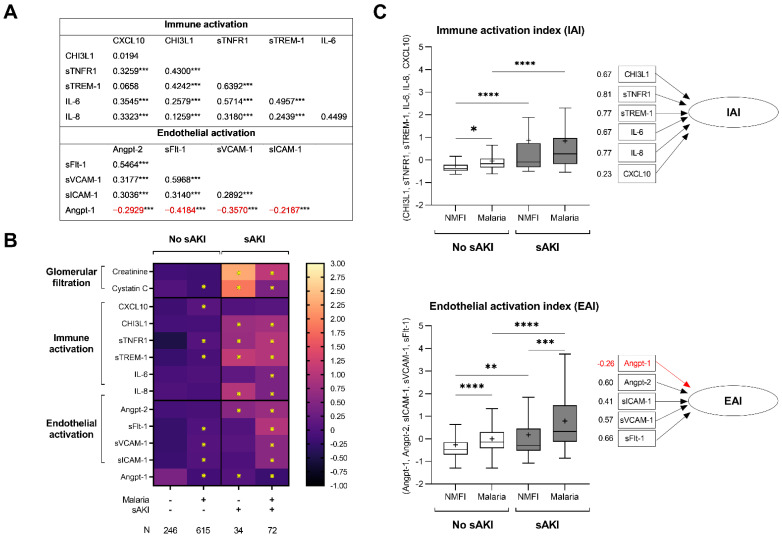
**Relationship between host markers of kidney injury and differences in pathways of immune and endothelial activation.** (**A**) Correlation matrix comparing biomarkers of immune activation and endothelial activation in the study population using Spearman rank correlation with Table presenting the rho value, and *** *p* < 0.001. Negative correlations are indicated in red. (**B**) Heatmap presents the mean standardized biomarker concentrations by malaria and AKI status (severe AKI (sAKI) vs. no severe AKI) with the number per group indicated. Biomarkers were categorized based on known biological function as markers of kidney function (glomerular filtration) or immune activation. Differences between groups were analyzed using linear regression with the standardized biomarker concentrations as the dependent variable and malaria and sAKI included as independent variables with an interaction term. The reference category included children without malaria (non-malarial febrile illness, NMFI) or severe AKI (sAKI, stage 2 or 3 AKI) and differences that were significant after adjusting for 36 multiple comparisons using the method of Benjamini–Hochberg false discovery rate were indicated by an asterisk. (**C**) Confirmatory factor analysis was used to derive an immune activation index (IAI) and an endothelial activation index (EAI). Differences in the indices were evaluated in children based on malaria and sAKI status using a one-way ANOVA with Sidak’s multiple comparison test where * adjusted *p* < 0.05, ** adjusted *p* < 0.01, **** adjusted *p* < 0.0001. The factor loadings for single principal components factor model with orthogonal rotation are presented beneath the Tukey’s boxplots with the mean presented (+).

**Table 1 pathogens-11-00436-t001:** Description of study population.

	*N*	Cohort (*n* = 967)	NMFI (*n* = 280)	Malaria (*n* = 687)	*p*-Value
**Demographics**					
Age, years	964	1.7 (1.1)	1.6 (1.1)	1.7 (1.1)	0.065
Female sex, n (%)	957	428 (44.7)	129 (46.2)	299 (44.1)	0.55
Weight, kg	963	9.8 (3.1)	9.5 (2.8)	9.9 (3.2)	0.055
Height, cm	948	74.0 (11.6)	72.8 (11.8)	74.5 (11.4)	0.034
**Medication history**					
Antimalarial n (%)	956	444 (46.4)	145 (52.2)	299 (44.1)	0.023
Antibiotic, n (%)	954	325 (34.1)	114 (41.0)	211 (31.2)	0.004
**Infection status**					
HIV, n (%)	966	20 (2.1)	12 (4.3)	8 (1.2)	0.002
**Clinical signs and symptoms**					
Axillary Temperature in °C	954	37.9 (1.2)	38.0 (1.1)	37.8 (1.1)	0.05
Systolic Blood Pressure, mmHg	931	105 (16)	104 (16)	105 (15)	0.14
Diastolic Blood Pressure, mmHg	929	58 (13)	58 (13)	57 (13)	0.35
Heart Rate	960	160 (25)	156 (25)	162 (24)	0.0009
Respiratory Rate	928	46 (15)	46 (15)	45 (14)	0.52
SpO2 %	960	96.8 (5.2)	95.9 (5.8)	97.2 (4.9)	0.0003
Capillary refill time > 2 s, n (%)	940	130 (13.8)	33 (11.9)	97 (14.6)	0.27
Unable to drink or breastfeed, n (%)	961	184 (19.2)	49 (17.6)	135 (19.8)	0.45
Vomiting, n (%)	963	293 (30.4)	87 (31.2)	206 (30.1)	0.74
Diarrhea, n (%)	964	289 (30.0)	108 (38.9)	181 (26.4)	<0.0001
Respiratory distress, n (%)	967	309 (32.0)	88 (31.4)	221 (32.2)	0.82
Prostration, n (%)	964	219 (22.7)	52 (18.6)	167 (24.4)	0.049
Coma (BCS < 3) n (%)	949	49 (5.2)	9 (3.3)	40 (5.9)	0.09
Altered consciousness, n (%)	961	134 (13.9)	30 (10.7)	104 (15.3)	0.064
Convulsions, n (%)	966	170 (17.6)	41 (14.6)	129 (18.8)	0.12
Jaundice, n (%)	966	104 (10.8)	16 (5.7)	88 (12.8)	0.001
Severe anemia (Hb < 5.0 g/dL), n (%)					
No Yes Missing	967	182 (18.8) 203 (21.0) 582 (60.2)	43 (15.4) 40 (14.3) 197 (70.4)	139 (20.2) 163 (23.7) 385 (56.0)	<0.0001
AKI, n (%)	967	475 (49.1)	154 (55.0)	321 (46.7)	0.02
Severe AKI (Stage 2 or 3), n (%)	967	106 (11.0)	34 (12.1)	72 (10.5)	0.45
Positive Cystatin C (≥0.8 mg/L), n (%)	967	188 (19.4)	73 (26.1)	115 (16.7)	<0.001
LOD Score, n (%)					
0 1 2 3	965	614 (63.6) 180 (18.7) 109 (11.3) 62 (6.4)	184 (65.7) 58 (20.7) 28 (10.0) 10 (3.6)	430 (62.8) 122 (17.8) 81 (11.8) 52 (7.6)	0.08
**Outcome**					
Death, n (%)	966	28 (2.9)	12 (4.3)	16 (2.3)	0.10

Data presented as mean (SD) or n (%).

**Table 2 pathogens-11-00436-t002:** Differences in disease presentation in children with malarial vs. non-malarial febrile illness based on AKI status at enrollment.

	No severe AKI (*n* = 861)			Severe AKI (Stage 2 or 3) (*n* = 106)		
	NMFI (*n* = 246)	Malaria (*n* = 615)	*p*-Value	NMFI (*n* = 34)	Malaria (*n* = 72)	*p*-Value
**Demographics**						
Age, years	1.6 (1.1)	1.7 (1.1)	0.12	1.7 (0.8)	2.0 (1.0)	0.16
Female sex, n (%)	115 (46.9)	278 (45.7)	0.75	14 (41.2)	21 (30.1)	0.26
Weight, kg	9.4 (2.8)	9.8 (3.2)	0.13	9.7 (2.7)	10.8 (3.1)	0.09
Length, cm	72.3 (11.8)	74 (11.3)	0.04	76.6 (11.2)	78.6 (12.1)	0.43
**Medication history**						
Antimalarial n (%)	126 (51.6)	267 (43.9)	0.04	19 (55.9)	32 (45.7)	0.33
Antibiotic, n (%)	95 (38.9)	190 (31.4)	0.03	19 (55.9)	21 (30.0)	0.01
**Infection status**						
HIV, n (%)	10 (4.1)	6 (0.9)	0.002	2 (5.9)	2 (2.8)	0.43
**Clinical signs and symptoms**						
Axillary Temperature in °C	37.9 (1.1)	37.9 (1.2)	0.14	37.9 (1.2)	37.5 (1.1)	0.07
Systolic Blood Pressure, mmHg	104 (16.8)	105 (15.1)	0.15	104 (13.5)	105 (16.3)	0.83
Diastolic Blood Pressure, mmHg	58 (13.3)	58 (13.2)	0.37	56 (10.9)	55 (12.0)	0.61
Heart Rate	156 (25.2)	162 (24.2)	0.0003	158 (22.9)	155 (25.4)	0.66
Respiratory Rate	46 (15.1)	45 (14.2)	0.42	45 (12.7)	45 (16.2)	0.81
SpO2 %	96 (6.1)	97 (5.0)	0.0002	97 (2.5)	97 (3.5)	0.91
Capillary refill time > 2 s, n (%)	27 (11.1)	81 (13.6)	0.31	6 (18.2)	16 (23.2)	0.57
Unable to drink/breastfeed, n (%)	39 (15.9)	114 (18.6)	0.36	10 (29.4)	21 (29.6)	0.97
Vomiting, n (%)	72 (29.4)	173 (28.2)	0.73	15 (44.1)	33 (46.5)	0.82
Diarrhea, n (%)	96 (39.2)	163 (26.6)	<0.0001	12 (36.4)	18 (25.0)	0.23
Respiratory distress, n (%)	75 (30.5)	188 (30.6)	0.98	13 (38.2)	33 (45.8)	0.46
Prostration, n (%)	41 (16.7)	133 (21.7)	0.10	11 (32.4)	34 (48.6)	0.12
Coma (BCS < 3) n (%)	7 (2.9)	27 (4.5)	0.28	2 (6.3)	13 (18.8)	0.10
Altered consciousness, n (%)	21 (8.5)	82 (133.4)	0.05	9 (26.5)	22 (30.9)	0.64
Convulsions, n (%)	33 (13.4)	121 (19.7)	0.03	8 (23.5)	8 (11.1)	0.10
Jaundice, n (%)	10 (4.1)	66 (10.8)	0.002	6 (17.7)	22 (30.6)	0.16
Severe anemia ^1^, n (%)						
No Yes Missing	40 (16.3) 32 (13.0) 174 (70.7)	121 (19.7) 136 (22.1) 358 (58.2)	0.001	3 (8.8) 8 (23.5) 23 (67.7)	18 (25.0) 27 (37.5) 27 (37.5)	0.01
LOD Score, n (%)						
0 1 2 3	164 (66.7) 54 (21.9) 21 (8.5) 7 (2.9)	400 (65.6) 109 (17.5) 71 (11.6) 34 (5.5)	0.12	20 (58.8) 4 (11.8) 7 (20.6) 3 (8.8)	30 (42.3) 13 (18.3) 10 (14.1) 18 (25.4)	0.13
**Outcome**						
Death, n (%)	9 (3.7)	7 (1.1)	0.01	3 (8.8)	9 (12.5)	0.58

Data presented as mean (SD) or n (%). ^1^ Hemoglobin < 5 g/dL.

## Data Availability

Deidentified data and a data dictionary for the specified data set will be available from the study principal investigator on reasonable request (kevin.kain@uhn.on.ca).

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
