# Peer review of "Pathophysiology of Acute Kidney Injury in Malaria and Non-Malarial Febrile Illness: A Prospective Cohort Study"

_pathogens, 2022, doi:10.3390/pathogens11040436_

Round 1

Reviewer 1 Report

Hawkes MT and colleagues investigated a possible correlation between immune and endothelial activation and AKI in a pediatric population in Uganda. The cohort study included 967 children with malaria or non-malarial febrile illness. The study has been clearly designed and the paper has been well written. 

Author Response

Thank you for your review and comments. 

Reviewer 2 Report

In the present study, Hawkes et al. investigated whether immune and endothelial activations are involved in the development acute kidney injury (AKI). Authors identified several biomarkers of immune (IL-8, sTNFR1, sTREM-1, CHI3L1) and endothelial (Angpt-2, sFlt-1) activation associated with AKI in children with both malaria and non-malarial febrile illness (NMFI). Moreover, they found the relation between endothelial activation and severe AKI in children with malaria. The premise and results in this study is interesting and warranted. However, this reviewer has some comments.

In this manuscript, “no severe AKI” group is consisted with no AKI samples and stage 1 AKI samples (see Table 1, Table 2, Figure 2, and Figure 3). “no AKI samples” and “stage 1 AKI samples” should be separated and analyzed each other. This reviewer considers that analysis using stage 1 AKI samples may lead to the discovery of new markers that predict the severity of the condition. Authors should investigate “stage 1 AKI samples” and compare with “no AKI samples” and “severe AKI sample”, respectively.

  1. Overall, the results and discussion suffer from over-interpretation. These sections need to be considerably improved.
  2. Lines 48-49: Improve the writing; the sentences are difficult to interpret.
  3. Line 64: Introduce animal studies (For example, Int J Mol Sci. 2021 Apr 19;22(8):4209. doi: 10.3390/ijms22084209.).
  4. Table 1 and 2: No table legend. Authors should describe details of tables in legend, such as parenthesis and superscript.
  5. Lines 90-91: Over-interpretation. Authors found that high frequency of jaundice was observed in severe AKI patients, but it is unclear whether jaundice is associated with severe AKI in patients with malaria.
  6. Lines 146 and 148, compared to: Lower ? Higher? Please clarify this sentence.
  7. Figure 2: No tab for malaria, such as “Nosevere AKI (615)”, “severe AKI (72), “” P value” and “RR”.
  8. Figure 2: No data of Angpt-2 in No severe AKI of NMFI. Moreover, parenthesis should be explained, For example, (4.1, 11.1). Are 4.1 and 11.1 mean lowest and highest value, respectively?
  9. Lines 202-203: In Fig.3B, it is seemed that endothelial activations are same levels between “no AKI sample” and “severe AKI” sample of NMFI. Moreover, the association remains unclear in this study. Improve interpretation and writing.
  10. Line 204, “AKI is a risk factor for mortality”: Improve interpretation. Malaria-associated AKI but not AKI.
  11. Lines 210-211, no severe AKI: Improve the writing. “no severe AKI” should be replaced with “Stage I AKI”.
  12. Lines 221-222: Improve the writing; the sentence is not completed.
  13. Lines 248-250: Improve the writing; the sentences are difficult to interpret. Moreover, Over-interpretation. Your data suggest that endothelial activation in AKI may be induced by increased circulating levels of cell surface molecules sICAM-1, sVCAM-1, and sFlt-1. Please improve the writing.
  14. Lines 290-291, NMFI with incidental parasitemia: Patients showing incidental parasitemia is not NMFI. Please revise English.

Minor comments:

  1. Line 30, 29.1%: 29.0%
  2. Lines 155-156: please provide full names for sTNFR1, CHI3L1 and sTREM-1 in the Results section, in addition to the M&M section.
  3. Figure 3C: No dotted lines under the asterisks. Please add dotted lines.
  4. Line 207, 4.1%: 4.0%
  5. Line 209, 4.6%: 4.5%
  6. Line 229, our study: this study?
  7. Line 246, multi-system: Systemic?

Author Response

We have included our reply as an attachment. 

Reviewer 3 Report

In this study, authors demonstrated immune and endothelial activation in child patients with malarial and non-malarial acute febrile illness (NMFI) with severe acute kidney injury (AKI). Authors identified biomarkers of endothelial and immune activation in children with severe AKI including elevated IL-8, sTNFR1, CH13L1, sTREM-1 (endothelial), Angpt-2, sFlt-1, sICAM-1 and sVCAM-1 (immune). Authors demonstrated that endothelial activation and mortality is increased in children with malaria and severe AKI in comparison to those with NMFI and severe AKI, suggesting a role for endothelial activation in the pathophysiology of malarial AKI and shedding light upon understanding the pathogenesis of AKI. The significance of the study was strengthened by a large sample size with minimal age variation and even sex distribution. The study is well designed and does not require major changes.

Minor revisions

Lines 48-49: Rewrite sentence, language is not clear

Line 172: Correct “jury” for “injury”

Line 178-179: “immune activation” is repeated twice

Line 194: Add “with” between “children NMFI”

Author Response

Thank you very much for your review and comments. We have reviewed and corrected all the points raised in the revised manuscript. 

Round 2

Reviewer 2 Report

I thank the authors for taking the time to address comments and questions by the reviewers.

Minor comments.

Line 193, vs. AKI, 8.8%: vs. “severe” AKI, 8.8%.

Figure 1, 280 NMFI (29.0%): 280 NMFI (29.1%) because 280/967 is 29.1%.